# Undiagnosed Diabetes and Prediabetes in Patients with Chronic Coronary Syndromes—An Alarming Public Health Issue

**DOI:** 10.3390/jcm10091981

**Published:** 2021-05-05

**Authors:** Natalia Drobek, Paweł Sowa, Piotr Jankowski, Maciej Haberka, Zbigniew Gąsior, Dariusz Kosior, Danuta Czarnecka, Andrzej Pająk, Karolina Szostak-Janiak, Agnieszka Krzykwa, Małgorzata Setny, Paweł Kozieł, Marlena Paniczko, Jacek Jamiołkowski, Irina Kowalska, Karol Kamiński

**Affiliations:** 1Department of Population Medicine and Lifestyle Diseases Prevention, Medical University of Bialystok, 15-089 Białystok, Poland; natalia.drobek@umb.edu.pl (N.D.); mailtosowa@gmail.com (P.S.); m.paniczko@gmail.com (M.P.); jacek909@wp.pl (J.J.); 2Department of Cardiology, University Hospital of Bialystok, 15-276 Białystok, Poland; 3Polish Mother’s Memorial Hospital Research Institute, 93-338 Łódź, Poland; piotrjankowski@interia.pl; 4Institute of Cardiology, Jagiellonian University Medical College, 31-008 Kraków, Poland; dczarnecka@interia.pl (D.C.); tragez88@wp.pl (P.K.); 5Department of Cardiology, School of Health Sciences, Medical University of Silesia, 40-055 Katowice, Poland; mhaberka@op.pl (M.H.); zgasior@ptkardio.pl (Z.G.); karolinaszostak@gmail.com (K.S.-J.); 6Mossakowski Medical Research Centre, Polish Academy of Sciences, 01-224 Warsaw, Poland; Dariusz.Kosior@cskmswia.pl; 7Department of Cardiology and Hypertension with the Electrophysiological Lab, Central Clinical Hospital the Ministry of the Interior and Administration, 00-124 Warsaw, Poland; pawlakagnieszka@interia.pl (A.K.); malgoskub@gmail.com (M.S.); 8Department of Clinical Epidemiology and Population Studies, Institute of Public Health, Jagiellonian University Medical College, 31-008 Krakow, Poland; apajak@neostrada.pl; 9Department of Internal Medicine and Metabolic Diseases, Medical University of Białystok, 15-089 Białystok, Poland; irinak@poczta.onet.pl

**Keywords:** diabetes, prediabetes, chronic coronary syndrome, Body Mass Index

## Abstract

Dysglycemia is a public health challenge for the coming decades, especially in patients with chronic coronary syndromes (CCS). We want to assess the prevalence of undiagnosed diabetes mellitus (DM) and prediabetes, as well as identify factors associated with the development of dysglycaemia in patients with CCS. In total, 1233 study participants (mean age 69 ± 9 years), who, between 6 and 18 months earlier were hospitalized for acute coronary syndrome or elective revascularization, were examined (71.4% men). The diagnosis of DM, impaired fasting glucose (IFG), impaired glucose tolerance (IGT) have been made according to World Health Organization (WHO) criteria. Based on the oral glucose tolerance test (OGTT) results, DM has been newly diagnosed in 28 (5.1%, mean age 69.9 ± 8.4 years) patients, 75% were male (*n* = 21). Prediabetes has been observed in 395 (72.3%) cases. IFG was found in 234 (42.9%) subjects, 161 (29.5%) individuals had IGT. According to multinomial logistic regression, body mass index (BMI) and high-density lipoprotein cholesterol (HDL-C) should be considered when assessing risk of development of dysglycaemia after discharge from the hospital. Among people with previously diagnosed DM, a significantly higher percentage were willing to change their lifestyles after the index event compared to other patients. Patients with chronic coronary syndromes suffer a very high frequency of dysglycaemia. Most patients with chronic coronary syndromes, especially those with high BMI or low HDL-C, should be considered for screening for dysglycemia using OGTT within the first year after hospitalization. A higher percentage of patients who were aware of their diabetic status changed their lifestyles, which added the benefit of timely diagnosis and treatment of diabetes.

## 1. Introduction

Diabetes mellitus (DM) has been called a public health challenge for the coming decades. Many epidemiological studies revealed that patients with DM had significantly higher risk of cardiovascular diseases, cardiovascular death and neurodegenerative disorders. Both atherosclerosis and glucose metabolism disorders have similar risk factors [1]. DM is a group of metabolic disorders characterized by increased levels of blood glucose, resulting from defects in insulin secretion or insulin action, or both. DM leads over time to notable damage to the vessels, heart, eyes, kidneys or nerves. Moreover, the risk of cardiovascular disease (CVD) is doubled in DM [2] and about 75% of deaths in people with diabetes are due to coronary artery disease [3]. The close association between DM and coronary atherosclerosis have been previously described [4].

The health consequences of prediabetes include not only progression to DM but also a higher risk of atherosclerotic cardiovascular disease and mortality after adjusting for multiple risk factors [5,6]. In a general population, impaired glucose tolerance (IGT) is related to a higher risk of all-cause mortality, coronary heart disease, and stroke [5]. According to an expert panel of the American Diabetes Association, up to 70% of individuals with prediabetes will eventually develop diabetes [7]. However, scientists have not yet come to an unanimous opinion whether treatment should always be applied in patients with prediabetes [8].

Type 2 DM is increasingly common primarily because of increased prevalence of the main non-genetic determinants of the disease like obesity and physical inactivity [9]. The long-term trend of increasing prevalence of obesity among patients with coronary artery disease has been shown recently [10]. According to the World Health Organization (WHO), 422 million people worldwide have diabetes [11]. In recent decades, the prevalence of type 2 DM has risen rapidly, particularly in developing countries [11]. Referring to national figures presents a worrying situation: in Poland, more than 3.5 million people (~9.5% of the population) suffer from DM; however, approximately 750.000 of these individuals have not been diagnosed yet [12]. According to the most recent report of the National Institute of Public Health in Poland from 2017 2,521,000 people were treated for DM, about 8% of the adult Polish population. This means that every 12th adult in Poland suffers from this disease while in a group aged 65 and over it is almost every fourth person (23%). Diabetes was more often diagnosed and treated in women (they accounted for 55% of all cases). Another study suggests that about 20% of people with diabetes do not know that they suffer from this disease [13]. Due to high incidence and morbidity of DM healthcare services’ budgets in Europe are strained. The average cost of treating a diabetic patient is almost double of a non-diabetic patient [13]. Diabetes and its complications are a serious public health problem, due to the prevalence, healthcare cost and mortality. DM and prediabetes problems affect millions of people around the world. This situation is further deteriorated by co-existing CVD, which remains the leading cause of death both in Poland and Europe [12,14]. Hence, patients who suffer from CVD and are at risk of DM or prediabetes should be under watchful medical supervision, more emphasis should be put on secondary prevention and timely diagnosis of dysglycaemia as early recognition and treatment of diabetes may significantly improve their prognosis [15]. 

The aim of the study was to assess the prevalence of undiagnosed DM and prediabetes in patients with chronic coronary syndromes (CCS), as well as identify factors associated with the development of dysglycaemia in patients with CCS. 

## 2. Materials and Methods

The study was a part of the POLASPIRE study [16], a multicenter cross-sectional study of patients with CCS. POLASPIRE is a federation of small regional observational studies sharing a common protocol and participating in the European Action on Secondary Prevention through Intervention to Reduce Events (EUROASPIRE V) programme and were conducted in four regions: in the northern part of Poland, in central Poland and in two southern regions. The design and method of the study were described previously [16]. A retrospective identification of patients who were discharged from one of the 14th hospitals who after invitation for a prospective visit were examined in leading centers in 4 geographical regions. All patients (≥18 years and <80 years of age at the time of their index event or procedure) invited to visit were hospitalized at least than 6 months ago in one of the departments due to: acute myocardial infarction, unstable angina, elective percutaneous coronary intervention (PCI) or coronary artery bypass surgery (CABG). Each part of the study was conducted by researchers who were previously trained. Retrospective data were obtained from medical records. The prospective visit considered of an interview with the patient using EUROASPIRE V questionnaire. Likewise, the following measurements were taken during the visit: height, weight, waist circumference and mean value from two blood pressure measurements. Peripheral intravenous fasting blood samples were collected at the time of the visit. Glycated haemoglobin (HbA1C), total cholesterol (TCH), high-density lipoprotein cholesterol (HDL-C), triglycerides (TG) and low-density lipoprotein cholesterol (LDL-C) have been measured in fasting venous blood. TCH, HDL-C and TG were analyzed in serum, and HbA1C in whole blood by the high-performance liquid chromatography (HPLC) method on a BIO-RAD D-10 (Hercules, CA, USA) analyzer (the reference range is <6%). Using the Friedewald formula including lipid profile, the LDL-C fraction was computed unless TG were elevated >4.5 mmol/L. The study protocol regarding an assessment of glucose metabolism was the same for all the patients. Glucose level was determined with the reference enzymatic method with hexokinase in each local laboratory. The diagnoses of DM, IFG or IGT were made according to WHO criteria (Appendix A). For people who did not declare diabetes or were taking metformin for other reasons, the standard oral glucose tolerance test (OGTT) after 75 g of glucose load was performed (*n* = 546). The body mass index (BMI) is defined as the body mass divided by the square of the body height (kg/m^2^) and commonly accepted BMI ranges are 25 to 30 for overweight and over 30 for obese [17]. Another parameter of obesity that was measured in the study is waist circumference with normal values in women <80 cm and in men <94 cm based on European population [17]. Changes in dietary habits and lifestyle modification were self-reported by a standardized interview questionnaire (EUROASPIRE V).

Statistical analysis was performed using STATA 16 (College Station, TX, USA) and IBM SPSS 25 (NY, USA). The variables were tested for normality using the Shapiro–Wilk test and compared between the categories of diagnosis based on Kruskall­–Wallis’ tests for ordinal or quantitative variables and the chi-squared test for nominal variables. Exact comparisons between specific groups were made with Dunn–Bonferroni post hoc tests. The criterion for statistical significance was set at *p* < 0.05 (confidence interval = 95%). To analyze the ROC (receiver operating characteristic) curve a binary variable were created where prediabetes and diabetes were assigned a value of 1 and normoglycemia a value of 0. 

## 3. Results

In total, 1233 study participants (mean age 69.9 ± 8.4 years, median 68) were examined (71% men). After exclusion of patients for whom we did not have any information concerning diabetic status or fasting glucose (*n* = 121), the examined group was classified into smaller subgroups according to the following criteria: (1) Individuals with previously diagnosed DM: on the basis of information from the hospital discharge letter or data collected during an interview, i.e., taking hypoglycemic drugs (excluding metformin given for non-diabetic reasons) or information about diabetes diagnosis provided by medical personnel in the past. This group was additionally divided into subgroups with diagnosed diabetes before the hospital admission or during the year between admission and enrollment in the study. (2) Participants without a history of DM: the majority had OGTT performed (*n* = 546) and the minority (*n* = 87) did not have OGTT due to the patients’ refusal and technical reasons. The groups are depicted in Figure 1. 

OGTT was performed in 546 individuals who did not self-report history of DM or use of any glucose-lowering medication (10 patients reported taking metformin for other reasons and they also did not have OGTT). Based on the OGTT results, DM has been diagnosed in 28 (5.1%) subjects: the newly diagnosed diabetes cases, of whom 21 (75%) were males. The prevalence of prediabetes in the examined population was observed in 395 (72.3%) cases. Isolated impaired fasting glucose (IFG) was found in 234 (42.9%) subjects and 161 (29.5%) individuals had IGT. Moreover, we have calculated the percentage of people with fasting blood glucose within the normal range or with solely IFG who have been diagnosed with DM (*n* = 17; 3.11%) or IGT (*n* = 135; 24.7%). We have summarized these results in Appendix A. It is worth pointing out that some of the patients with fasting glucose ≥26 mg/dL might still fulfill diagnostic criteria for diabetes if there was a second fasting measurement available. Thus the number of patients with CCS who develop diabetes might be even higher.

Table 1 provides a comparison of the characteristics of patients with DM diagnosed before the enrolment to the current study with patients without diagnosed diabetes. The latter group was divided by the presence of OGTT test during the enrolment visit. DM and non-DM groups significantly differed in lifestyle changes in the period after hospitalization—a greater percentage of individuals with previously recognized DM decided to reduce the consumption of fats as well as sugar and alcohol. Subjects with previously diagnosed diabetes were considerably older than those without and also had on average higher body weight, waist circumference, BMI and HbA1C than other patients. On the other hand, they were characterized by better lipid profile. 

Table 2 and Table 3 contain the characteristics of such patients with their categorization according to the diagnosis. These subgroups did not differ in regard to gender distribution, index event and variables characterizing lifestyle changes in the period after hospitalization. Individuals with newly diagnosed DM declared that they did not reduce their dietary sugar intake more frequently than patients with other diagnoses. Statistically relevant differences between patients with various types of diagnosis were noted: people with IGT and DM were older than those with IFG. Moreover, patients with normal glucose tolerance (NGT) had significantly lower body weight and BMI than IGT and IFG. Dissimilarities between NGT and newly diagnosed DM were also observed for waist circumference, BMI and HbA1C, which were significantly different in all subgroups. People with pre-diabetes (IFG or IGT) had significantly higher BMI than those with normoglycemia. Such patients were also characterized by a larger waist circumference. However, no significant differences were found for blood lipids. 

Between the discharge from hospital and enrolment to the study, diabetes was diagnosed in 27 patients; OGTT performed in the study diagnosed DM in next 28 participants—altogether 55 patients out of 573 (9.6%) developed DM during the first year after the hospitalization due to CVD. In our study another group of 161 people presented with IGT. According to the current indications, they should already be the subject of active diabetes prevention (lifestyle change-physical activity, healthy eating, eventually metformin treatment). 

Table 4 and Table 5 summarize the characteristics of all patients with clinical diagnosis of DM (*n* = 452) prior to the examination, of 27 patients with DM diagnosed after hospitalization and of 28 newly diagnosed during the POLASPIRE study. The studied subgroups did not differ in terms of anthropometric characteristics (BMI, body weight, waist circumference) and lipid management. Significant differences in HbA1C were observed between patients with DM already diagnosed before the hospitalization and the other DM patients. Therefore, it is challenging to identify parameters that might indicate that a patient hospitalized due to CVD may develop dysglycemia in the future.

In order to estimate possible predictors of dysglycemia development (dependent variable) in the period after the index event, multinomial logistic regression analyses were performed with one (Univariate) or multiple independent variables (Multivariate)—respectively, the results are listed in Appendix A. Univariate analysis allowed to identify variables that could significantly increase the odds of developing dysglycemia (*p* < 0.05). The multivariate model (independent variables: HDL-C, triglycerides, BMI) showed that the odds of developing prediabetes compared to staying in normoglycemia were 1.123 (confidence interval (CI): 1.042–1.209) times greater for each unit increase in BMI measured during hospitalization (index event). The model with independent variables of HDL-C and BMI indicated that the odds of developing prediabetes multiplies by 1.114 (CI: 1.035–1.198) for each increase of BMI units; in comparison to normoglycemia, the odds of developing DM are lower among patients with higher HDL-C concentration (odds ratio (OR) = 0.145, CI: 0.038–0.546). The analysis of the ROC curves (see Figure 2) confirmed that BMI during hospitalization was a good predictor of the development of dysglycemia (area under the curve (AUC) = 0.63; CI: 0.563–0.704). Additionally, according to the criterion of maximizing the Youden’s index (J Statistic Youden Index= 0.213), for a BMI of 29.73 the sensitivity is 0.407 and specificity is 0.806.

## 4. Discussion

This publication reveals the epidemiological situation of undiagnosed diabetes, especially in patients with CVD. Studies conducted in the general Polish population (NATPOL 2011) have shown that the prevalence of DM (both, diagnosed and undiagnosed) in the entire study population aged 18–79 was 6.7% [18]. Undiagnosed cases in this population were 1.9%, which is a smaller percentage than in our study population. In addition, a higher percentage of people with unrecognized diabetes were found in the group of men, both in our and NATPOL research [18]. Another population study (WOBASZ Surveys) confirmed the high rate of detected DM cases (8.4%) and the increased prevalence in the male population [13]. Moreover, IFG was found in 18.4% which also shows a very high incidence. In that paper population incidence of DM and IFG varies depending on age and sex, similarly to our study. Additionally, the data obtained indicated that over the last decade (2004–2014) in Poland the prevalence of diabetes has increased significantly from 6.6% to 8.4%. There are alarming data also regarding the people with IFG—an increase by as much as 50% (from 9.3% to 18.4%) has been reported [13]. According to the National Diabetes Statistics Report, in the United States, over 7 million Americans have undiagnosed diabetes accounting for 2.9% of the total population [19]. Other authors have estimated the global number of undetected DM in adults and referring to their results, approximately half of all DM cases (48.5%; from 24.1% to 75.1% across data regions) remain undiagnosed [20]. More recent studies in the Asian population also indicate that both DM and prediabetes are prevalent and often remain unrecognized [21]. Previous studies emphasize that the prevalence of undiagnosed disturbances in glucose metabolism is high worldwide, and the number of cases is increasing, which is an alarming public health challenge.

It is widely accepted that diabetic patients have higher risk of cardiovascular events. In our study we pointed out that the patients with history of CVD have high risk of developing DM. We present the view that patients who have had a cardiovascular intervention and have been previously hospitalized should be carefully screened for developing dysglycaemia and subsequently the therapy (pharmacological and non-pharmacological) should be initiated when necessary. In our work, the prevalence of dysglycaemia is very high despite recent medical consultation in hospital. The observation period may be too short to fully appreciate the factors involved in the process and further studies are needed to extend this period and suggest appropriate measures. Nevertheless, even in this short period of time we present an alarmingly high percentage of newly diagnosed diabetes and IGT, so we suggesting the necessity of frequent testing for glycemic control.

In this study, we found a very high prevalence of DM and prediabetic states in a group of patients with diagnosed CVD, especially with high BMI. This is in accordance with The Euro Heart Survey, which demonstrated that the abnormal glucose regulation is more common than normal glucose level in patients with CVD [22]. The study done in Germany has shown that in patients with hypertension (HT), BMI above the norm and >45 years the incidence of diabetes is higher than in patients who do not have concomitant diseases [23]. 

Our study confirms the conclusions of a paper published recently from the Euroaspire V database [24]. Despite clear guidelines for the diagnosis of glucose metabolism disorders, many people remain undiagnosed even with coronary heart disease [24]. Differences in the number of diagnoses may result from the methods of glucose determination, as well as differences between standards of care between various countries. Part of the population studied in our project was also included in the Euroaspire V registry. In our study, however, we used the reference laboratory methods in opposition to the point of care method used in the aforementioned study [24]. Additionally, in our study, we did not diagnose glucose disturbances based on HbA1C levels. In our report, we pointed out which parameters (BMI, HDL-C) should be considered when discharging from the hospital due to the increased probability of DM and what may be the consequences of delayed DM diagnosis on lifestyle modifications. It should be emphasized that the prevalence of dysglycemia is much higher in our CCS cohort in comparison to general population mainly due to the studied population characteristics-all study participants had significant coronary atherosclerosis. The contribution of insulin resistance/hyperinsulinemia and hyperglycemia in acceleration of the process of atherosclerosis is well documented [3,5]. When comparing other earlier studies in patients with CCS [4,21], our data present steady increase of prediabetes in time, simultaneously with obesity pandemic. Apparent drastic differences may be due to different definitions of prediabetes in our study (we included both IGT and IFG) and previous papers (earlier studies usually included exclusively IGT). 

A recent study has shown that there is negative correlation between HDL and BMI [25]. HDL-C was found to be significantly higher in diabetic patients with normal BMI [25]. In our study, we admittedly showed a negative correlation between these risk factors. However, our study emphasizes the diagnosis of dysglycaemia in patients with cardiovascular disease, which was not considered in that study. Another non-European study also found a weak negative correlation of HDL-C with BMI [26]. The correlation of these factors has been described previously by researchers, however they did not identify them as factors that we should particularly consider when discharging cardiac patients at risk of dysglycaemia from hospital. Additionally, we show that well-known factors should be used in clinical practice, in particular in patients with CCS.

We also observed that patients are often not aware that they have abnormal glucose plasma concentrations. Prior studies also have shown that undiagnosed DM, compared to diagnosed DM, was linked to significantly higher CVD risk and prone to uncontrolled HT and elevated LDL [27]. Likewise, undiagnosed DM has been reported to carry the same risk of mortality to diagnosed DM, but it is also associated with a higher risk of mortality compared to normoglycemic patients [28]. We show here that patients who were aware of their diabetic status were much more compliant with lifestyle modifications, hence providing further benefit of timely diabetes diagnosis.

The prevalence of prediabetes is increasing worldwide. Reports estimate that more than 470 million people will have prediabetes by 2030 [29]. Prior studies have shown that prediabetes was associated with an increased risk of mortality and cardiovascular disease in the general population and patients with CCS [5,6]. Scientists are considering whether inducing a treatment already at the stage of prediabetes is an appropriate solution, due to the fact that the moment of intermediate hyperglycemia transition to DM is missed and untreated for a long time. All of them agree with the emphasis on lifestyle changes in the form of regular physical activity and a healthy balanced diet. There are insufficient studies on the treatment of prediabetes, whether it brings more benefit or harm. According to the Diabetes Prevention Program, in patients with prediabetes life style changes reduce the occurrence of DM by 58%. Moreover, regular use of metformin (2 × 850 mg) reduces its occurrence by 31% during 2.8 years of follow-up [29]. However, future studies should consider more cardiovascular endpoints instead of concentrating solely on glucose control to properly assess the potential benefits of treatment of dysglycaemia. In conclusion, screening and proper management of prediabetes might contribute to the prevention of cardiovascular disease.

The economic analysis shows that the total costs related to type 2 DM in the US increased by 41% between 2007 and 2012 [30]. Prediabetes is associated with an escalation in the number of medical services, as well as an increase in drug and medical products expenditure [31,32]. Compared with patients who did not progress, the total adjusted medical costs for patients who developed diabetes increased by 26.7%, 24.2%, and 23.4% at 1.2 and 3 years after progression [33]. In conclusion, it should be beneficial to prevent progression and complications of prediabetes, as after a few years, such patients are a greater burden for the health system. Noting that the symptoms of DM develop insidiously, many cases remain undetected, even in countries with a well-developed healthcare infrastructure. Our study suggests that earlier diagnoses could lead to more proactive responses, better patient outcomes, and significant savings in healthcare costs–in particular for those individuals unaware that they currently have prediabetic symptoms. It is also advisable to consider earlier inclusion of pharmacological treatment, e.g., in patients with non-diabetic dysglycemia. 

Recently, Haberka et al., whose study was based on part of the cohort presented here, showed that the majority of Polish patients in secondary prevention do not achieve treatment goals [34]. It has been shown that patients with DM only in 60% reached the diabetic goal–more intensive treatment and its control is necessary.

The prevalence of glucose metabolism disorders is high among CVD patients with specific risk factors; therefore, a strong justification for the use of targeted diabetic screening such as this could reduce the occurrence of long-term complications and also reduce the risk of death. New and better screening would allow patients with prediabetes better glycaemic control and prevent or retard the onset of diabetes, especially in obese or after acute coronary syndromes [10,35,36]. In our model it seems that BMI is the main indicator we could use to predict the development of dysglycemia after discharge from the hospital in CVD patients.

### Strengths and Limitations of the Study

Our multicenter study provides data among Polish patients with a history of cardiovascular incidents and with or without diabetes. One of the strengths is that it was performed in cardiac departments of various references using a standardized interview. As a limitation one can consider that patients who died between hospitalization and the visit or those who did not report to the prospective examination could also have different results. In addition, some patients were unable to perform OGTT for technical reasons and were excluded from post-load glucose measurement; however, their characteristics did not differ from the patients who had OGTT performed. Therefore the estimated number of undiagnosed DM may be underestimated. Another limitation is the fact that many patients do not know their medical history, with what medications they are being treated for and whether they have the disease or not. 

## 5. Conclusions

The incidence of undiagnosed DM and prediabetes is very high in patients with CVD. The measurement of fasting glycaemia alone is not enough to diagnose DM and prediabetes, and it is necessary to perform OGTT in patients with a high risk of development of DM. Our results suggest that OGTT should be performed in all patients with CCS one year after hospitalisation. However, if there are limitations such as financial or technical, special attention should be paid to patients with high BMI and elevated HDL-C levels. Patients who were aware of their diabetic status were much more compliant with lifestyle modifications, hence providing further benefit of timely diabetes diagnosis; and, conversely, patients who do not comply with lifestyle modification, especially limitation of dietary simple carbohydrates, may have a higher risk of diagnosis of DM. 

## Figures and Tables

**Figure 1 jcm-10-01981-f001:**
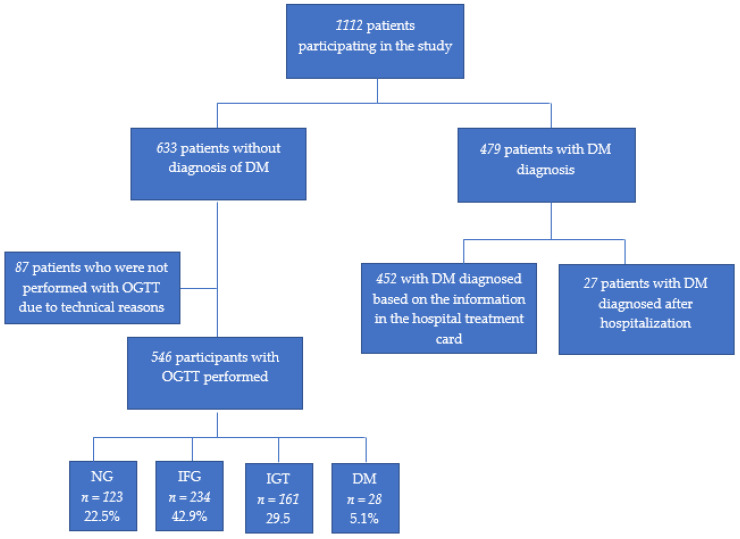
The division of participants into groups.

**Figure 2 jcm-10-01981-f002:**
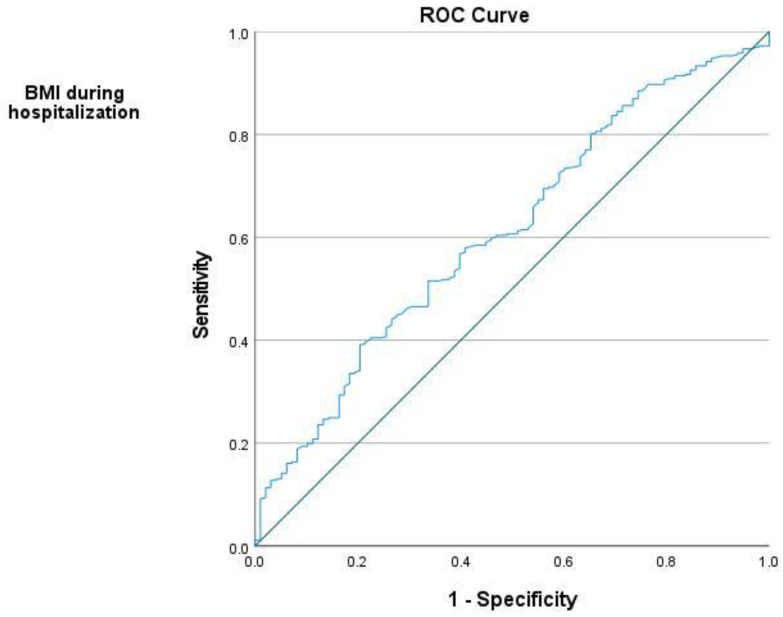
Receiver operating characteristic (ROC) curve (area under the curve (AUC) = 0.63; *p* = 0.001); larger values of the test result variable indicate stronger evidence for a positive actual state.

**Table 1 jcm-10-01981-t001:** Patients’ characteristics with diabetes mellitus (DM) diagnosed before participation in the POLASPIRE study with patients with and without oral glucose tolerance test (OGTT) performed.

		DM *n* = 47	IQR ^1^	No DM, OGTT Performed *n* = 546	IQR ^1^	No DM, OGTT Was Not Performed *n* = 87	IQR ^1^	*p*
Sex	M *n* (%)	328 (68.5)		399 (73.1%)		63 (72.4%)		0.258
Age median		70 ^a^	64.0–75.0	67 ^a^	62.0–73.0	67	63.0–75.0	<0.01
BMI (kg/m^2^) median		30.5 ^a,b^	27.5–33.6	28.64 ^a^	25.9–31.5	28.33 ^b^	25.3–30.5	<0.01
Body weight (kg) median		86.00 ^a,b^	77.0–97.0	81.28 ^a^	73.0–91.0	79.0 ^b^	71.0–90.0	<0.01
Waist circumference (cm) median		105.0 ^a,b^	98.0–113.0	100.0 ^a^	95.0–108.0	99 ^b^	90.0–104.0	<0.01
Total cholesterol (mg/dL) median		145.0 ^a,b^	123.4–177.9	153 ^a^	132.1–181.7	160 ^b^	141.5–193.3	<0.01
LDL (mg/dL) median		70.69 ^a,b^	54.1–92.8	80.72 ^a^	64.8–101.7	83.20 ^b^	66.0–109.3	<0.01
HDL-C (mg/dL L) median		46 ^a,b^	38.7–56.5	50 ^a^	42.9–60.5	50.66 ^b^	44.7–58.9	<0.01
TG (mg/dL) median		126.4 ^a,b^	89.5–169.2	105 ^a^	78.9–143.7	105.1 ^b^	74.0–138.0	<0.01
HbA1C (%) median		6.5 ^a,b^	6.0–7.4	5.70 ^a^	5.5–6.0	5.6 ^b^	5.4–5.9	<0.01
Index event *n* (%)	CABG	17 (3.5%)		23(4.2%)		6(6.9%)		<0.01
	PCI	185(38.6%) ^a^		208(38.1%) ^b^		12(13.8%) ^a,b^		
	STEMI	62(12.9%) ^a^		88(16.1%) ^b^		27(31%) ^a,b^		
	NSTEMI	121(25.3%)		105(19.2%)		24(27.6%)		
	Unstable angina	94(19.6%)		122(22.3%)		18(20.7%)		
lifestyle change *n* (%)	reduction in salt consumption	272(71.6%)		352(66.3%)		51(59.3%)		0.08
	reduction in fat consumption	303(79.5%) ^a,b^		365(67.8%) ^a^		55(64%) ^b^		<0.01
	reduction in calories consumption	250(65.8%) ^a^		323(60.3%)		38(44.2%) ^a^		<0.01
	more fruits and vegetables consumption	291(76.4%)		378(70.1%)		62(72.1%)		0.266
	more fishes consumption	174(45.7%)		240(44.8%)		41(47.7%)		0.610
	reduction in sugar consumption	307(80.6%) ^a,b^		320(59.5%) ^a^		48(55.8%) ^b^		<0.01
	reduction in alcohol consumption	247(65.2%) ^a,b^		299(55.8%) ^a^		38(44.2%) ^b^		0.03
	compliance with dietary recommendations	258(67.7%) ^a^		273(50.8%) ^a^		47(54,7%)		<0.01

^1^ IQR—Interquartile range; Significance differences obtained by Dunn–Bonferroni’s post hoc test at the 0.05 level; statistically significant pairs are marked with lowercase letters (a, b). BMI, body mass index; LDL, low-density lipoprotein; HDL-C, high-density lipoprotein cholesterol; TG, triglycerides; HbA1c, glycated haemoglobin; CABG, coronary artery bypass surgery; PCI, percutaneous coronary intervention; STEMI, ST-segment elevation myocardial infarction; NSTEMI, non-ST elevation myocardial infarction.

**Table 2 jcm-10-01981-t002:** Patients’ characteristics by glucose category in group with OGTT.

Categories of Diagnosis										
		NGT *n* = 123	IQR	IFG *n* = 234	IQR	IGT *n* = 161	IQR	Newly Diagnosed DM *n* = 28	IQR *	*p*
sex	M *n* (%)	85 (69.1%)		181 (77.4%)		112 (69.6%)		21 (75%)		0.239
Age median		67.0	60.0–73.0	65.0 ^a,b^	61.0–62.0	69.0 ^a^	63.0–75.0	69.0 ^b^	66.0–73.0	<0.05
BMI (kg/m^2^) median		27.5 ^a,b,c^	24.5–29.9	28.6 ^a^	25.9–31.5	29.4 ^b^	26.6–32.6	29.3 ^c^	27.2–30.7	<0.05
Body weight (kg) median		80.0 ^a,b^	67.0–88.5	82.0 ^b^	75.0–93.0	81.2 ^a^	74.0–92.2	81.0	74.3–90.5	<0.05
Waist circumference (cm) median		99.0 ^a,b,c^	92.0–105.0	100.0 ^a^	95.0–108.0	103.0 ^b^	95.0–110.0	102.5 ^c^	99.0–107.0	<0.01
Total cholesterol (mg/dL) median		155.5	134.0–187.0	150.8	131.5–180.0	154.0	132.0–181.5	144.5	132.5–170.0	
LDL (mg/dL) median		85.8	66.8–106.0	77.3	65.0–100.5	78.9	64.8–101.3	80.2	56.8–106.1	
HDL-C (mg/dL L) median		52.5	43.0–63.1	49	42.1–58.4	51.0	43.0–61.0	67.6	42.0–54.0	
TG (mg/dL) median		101.9	76.0–135.0	107.0	80.0–145.0	109.0	78.0–144.4	120.5	98.8–195.3	
HbA1C (%) median		5.6 ^a,b,c^	5.4–5.8	5.7 ^a,d,e^	5.5–5.9	5.8 ^b,d,f^	5.6–6.1	6.1 ^c,e,f^	5.7–6.4	<0.01

* IQR—Interquartile range; Significance differences obtained by Dunn–Bonferroni’s post hoc test at the 0.05 level; statistically significant pairs are marked with lowercase letters (a,b,c,…).

**Table 3 jcm-10-01981-t003:** Index event and lifestyle changes in groups of patients according to OGTT.

		NGT *n* = 123	IFG *n* = 234	IGT *n* = 161	Newly Diagnosed DM *n* = 28	*p*
**Indexe vent** ***n* (%)**	CABG	6(4.9%)	12(5.1%)	4(2.5%)	1(3.6%)	
	PCI	52(42.3%)	84(35.9%)	61(37.9%)	11(39.3%)	
	STEMI	22(17.9%)	38(16.2%)	21(13%)	7(25%)	0.579
	NSTEMI	19(15.4%)	50(21.4%)	34(21.1%)	2(7.1%)	
	Unstable angina	24(19.5%)	50(21.4%)	41(25.5%)	7(25%)	
**lifestyle change *n* (%)**	reduction in salt consumption	79(65.8%)	151(66.2%)	108(69.2%)	14(51.9%)	0.607
	reduction in fat consumption	83(67.5%)	156(68.1%)	112(70.9%)	14(50%)	0.228
	reduction in calories consumption	72(58.5%)	143(63%)	92(58.2%)	16(57.1%)	0.191
	more fruits and vegetables consumption	92(74.8%)	160(69.6%)	108(68.4%)	18(64.3%)	0.199
	more fishes consumption	52(42.3%)	98(43%)	77(49%)	13(46.4%)	0.534
	reduction in sugar consumption	74(60.7%) ^a^	137(59.6%) ^b^	100(63.3%) ^c^	9(32.1%) ^a,b,c^	0.034
	reduction in alcohol consumption	68(56.2%)	131(57.2%)	87(55.1%)	13(46.4%)	0.584
	compliance with dietary recommendations	56(45.5%)	117(51.3%)	84(53.2%)	16(57.1%)	0.787

Significance differences obtained by Dunn–Bonferroni’s post hoc test at the 0.05 level; statistically significant pairs are marked with lowercase letters (a, b, c). The distribution of diagnostic categories for males and females in patients with OGTT performed was not significantly different (*p* = 0.239) as presented in Appendix A.

**Table 4 jcm-10-01981-t004:** Characteristics of all patients with DM.

	Reported during Hospitalization *n* = 452	IQR	After Hospital Discharge *n* = 27	IQR	Newly Diagnosed *n* = 28	IQR	*p*
Sex	307 (67.9%)		21 (77.8%)		21 (75%)		0.432
Age median	70.0	65.0–75.0	64.0	61.0–74.0	70.0	66.0–73.0	0.053
BMI (kg/m^2^) median	30.6	27.4–33.5	29.4	27.4–35.6	30.6	27.2–30.7	0.260
Body weight (kg) median	86.0	77.0–96.5	85.9	76.6–99.5	86.0	74.3–90.5	0.264
Waist circumference (cm) median	105.0	98.0–113.0	103.0	99.0–120.0	105.0	99.0–107.0	0.450
Total cholesterol (mg/dL) median	145.0	123.7–177.8	143.3	123.7–170.0	145.0	132.5–170.0	0.886
LDL (mg/dL) median	70.6	54.1–93.9	71.7	54.1–83.0	70.6	56.8–106.1	0.468
HDL-C (mg/dL L) median	46.0	38.7–56.8	43.0	37.0–47.0	46.0	42.0–54.0	0.236
TG (mg/dL) median	124.9	89.0–166.0	168	93.0–213.3	124.9	98.8–195.3	0.119
HbA1C (%) median	6.6	6.0–7.5	6.0	5.8–6.8	6.6	5.7–6.4	<0.01

IQR—Interquartile range; Significance differences obtained by Dunn–Bonferroni’s post hoc test at the 0.05 level.

**Table 5 jcm-10-01981-t005:** Index event and life style changes of all patients with DM.

		Reported during Hospitalization *n* = 452	After Hospital Discharge *n* = 27	Newly Diagnosed *n* = 28	*p*
Index event *n* (%)	CABG	14(3.1%)	3(11.1%)	1(3.6%)	0.030
	PCI	179(39.6%)	6(22.2%)	11(39.3%)	
	STEMI	55(12.2%)	7(25.9)	7(25%)	
	NSTEMI	114(25.2%)	7(25.9%)	2(7.1%)	
	Unstable angina	90(19.9%)	4(14.8%)	7(25%)	
lifestyle change *n* (%)	reduction in salt consumption	255(72%)	17(65.4%)	14(51.9%)	0.131
	reduction in fat consumption	286(80.6%) ^a^	17(65.4%)	14(50%) ^a^	<0.01
	reduction in calories consumption	234(66.1%)	16(61.5%)	16(57.1%)	0.449
	more fruits and vegetables consumption	270(76.1%)	21(80.8%)	18(64.3%)	0.556
	more fishes consumption	165(46.5%)	9(34.6%)	13(46.4%)	0.748
	reduction in sugar consumption	290(81.7%) ^b^	17(65.4%) ^a^	9(32.1%) ^a,b^	<0.01
	reduction in alcohol consumption	230(65.2%)	17(65.4%)	13(46.4%)	0.378
	compliance with dietary recommendations	243(68.5%)	15(57.7%)	16(57.1%)	0.162

Significance differences obtained by Dunn–Bonferroni’s post hoc test at the 0.05 level; statistically significant pairs are marked with lowercase letters (a,b) 3.1. HDL-C and BMI as variables that increase the probability of developing DM after hospitalization.

## Data Availability

Data available on request only for scientific purposes. More details on https://bialystok.plus (accessed on 17 February 2021).

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
