# Peer review of "Undiagnosed Diabetes and Prediabetes in Patients with Chronic Coronary Syndromes—An Alarming Public Health Issue"

_jcm, 2021, doi:10.3390/jcm10091981_

Round 1

Reviewer 1 Report

Kamiński and colleagues assessed the prevalence of undiagnosed diabetes and prediabetes in patients with chronic coronary syndrome (CCS). Therefore, they conducted a multi-center national trail among 1233 patients with hospitalization due to acute coronary syndrome or elective revascularization 6 month after the index event. The authors concluded that screening for dysglycemia using OGTT is especially beneficial among CCS patients with high BMI and/or low HDL-C, whereas in patients with previously diagnosed DM, a higher willingness to change their lifestyle was observed after the index event compared to patients without previous diagnosis of DM.

Cardiovascular risk in patients with diabetes and prediabetes is a very interesting and a current research topic. However, there are some major concerns within the study at hand listed beneath. Especially the lack of a clear study intervention strategy and quality assessment limits the significance of the study and makes interpretation of results somewhat doubtful (treats to the internal validity). The study cohort is very interesting, but the study question per se remains unclear to this reader. A more comprehensive design with a clear research question could help to generate a more rigorous structure to this study. It would have been very interesting to see results of a in depth analysis of those patients that developed DM after the initial event.

Major Concerns:

  1. Kamiński et al. stated in the Methods section an initial drop-out rate of 10% (n=124) by missing information on DM and a secondary drop-out rate of 14% (n=87) by failing to test for DM. This seems like a non-neglectable fraction of the study population. The authors should consider potential confounding of results and discuss this in their manuscript in more detail.
  2. The authors should be careful regarding the following confusing aspects:
    • They initially stated that the want to explore the prevalence of undiagnosed diabetes and prediabetes in patients with CCS. Later they authors interpret results of an intervention and comment on lifestyle changes and differences in eating habits. The study investigators should clarify their research question/purpose in this study. This would help the reader to follow the manuscript and the thesis within much better.
    • In line 328 the researchers draw conclusions that are not reflected in their data within the study, such as that fasting glucose testing is not enough for risk stratification. This statement might be true, but there is no evidence supplied by the data at hand, that justify the conclusion.
    • KamiĹ„ski et al. should consider that the prevalence of events in the study cohort might be elevated, but that the observation period might still be too short to fully depict difference in specific subgroups. Especially since all patients have a certain risk exposure, since they already established cardiovascular disease. On that note, the duration of the follow-up period was not clearly stated to the best of my knowledge.
    • The authors should be careful when interpreting their results, especially with small subgroups, such as newly diagnosed DM (n=28). Moreover, the differences indicated by the ROC statistics for BMI are slightly above the line of indifference and failed to provide the confidence base that should be within the data. A sensitivity of 40,7% at the calculated BMI cut-off point does not strengthen the confidence.
    • It remains unclear to the reader what the quality indicators of the study intervention were? How were the lifestyle changes measured? How was a change in eating habits verified?
  3. The major concern rises by the comparing the results at hand with previous trails such as ARTEMIS and EUROASPIRE IV. The study investigators should elaborate on the novelty that their study has and what separates their results from previous publications in the field.
  4. Moreover, the meaningfulness of the results should be discussed in more detail and set into context with the existing evidence base. That the prevalence of DM/prediabetes is correlated with BMI and HDL-C is not new. The authors should expand on their thoughts that their results add value to the current evidence base.
  5. The Authors should clearly state what the intended study intervention was? Collecting data by a questionary does not qualify as an intervention. Neither does the index event. What was the control group?
  6. A very interesting aspect of the findings in this study is that they identified predominately male patients with impaired glucose tolerance. This seems to contradict common evidence in the field, what should be elaborated in-deth by the researchers. One aspect of that could be that the majority of patients in the study cohort were male and that it does not represent an unambiguous sample.
  7. The Authors should adept IGT/IFG to ESC rather than WHO guidelines/definition
  8. Acute myocardial infarction has a major impact on the metabolism. Is it possible to distinguish between those patients that had elective PCI and the ones with ACS? Would be very interesting to investigate for differences in these two groups. The authors should elaborate on this very subgroup analysis in particular.

Minor Concerns:

  1. There are several double blank mistakes within the manuscript that should be revised. Please refer lines 45, 46, 50, …
  2. Prospective trails should be registered and evaluated prior to their start. The investigators failed to register POLASPIRE prospectively on clinicaltrails.gov.
  3. Tables 2 and 3 are very hard to comprehend. The format should be updated. There is just one p value displayed for three groups in Table 1 and it is unclear to which groups it displays the statistical difference. The 3rd group in Table 2 without DM and without a OGTT test should be abolished since it adds no value to the understanding of the subgroups.
  4. The subgroup of patients with newly diagnosed DM after discharge should be exploited more. Did they already have an impaired OGTT? What are the risk factors among those individuals?

Reviewer 2 Report

This is an interesting epidemological study showing a high prevalence of diabetes and prediabetes in patients with CCS in Poland.

Of all 1112 study participants, 546 had an OGTT, and from those IGT was diagnosed in almost 30% and DM in 5.1%, which is pretty high and deserves attention. High BMI and low HDL predicted diabetes during follow-up, which is not a very novel finding.

The authors should adress the following issues:

  1. The conclusion that fasting glycemia alone is not enough to diagnose DM/PD cannot be concluded from this study design, since fasting glycemia was not routinely available. Please rephrase.
  2. The placement of tables and figures is not appropriate in the results section. Please re-organize and make it clearer to the reader. Also the dicussion needs rearrangement. The authors should start with their major findings and what is novel in their study.
  3. What were reasons not to perform OGTT? Was there any dysbalance between those and those who had OGTT?
  4. Why was elevated HbA1c not an inclusion / exclusion criterion to diagnose prediabetes /diabetes?
  5. The impact of having DM vs. not having it on lifestyle is not clearly presented in the results section
  6. Table 4: what statistical test was used for comparison (of more than 2 groups)? ANOVA? With our without post-hoc correction?
  7. What is the final message of the authors? OGTT in all patients with CCS not having DM? What to do with the obese?

Reviewer 3 Report

Summary:  Data from a multicenter study of Polish patients with chronic coronary syndrome (CCS) evaluating the incidence of undiagnosed diabetes and pre-diabetes are presented.  Not surprisingly, given their median age and BMI (70 years and 30.5 kg/m2), there was a high incidence of undiagnosed diabetes and a very high prevalence of pre-diabetes.  It is interesting that a much higher fraction of the pre-diabetes cohort was composed of patients with impaired glucose tolerance (IGT) only compared to other studies of pre-diabetes incidence in general populations. This emphasizes the importance of applying OGTT to this population to identify those at risk of developing diabetes.  It would be important to determine if such an unusual distribution of IGT/IFG has been previously reported for pre-diabetes patients with underlying chronic heart disease.

Major Points: 

  1. For the CCS cohort, the proportion of pre-diabetes subjects with IGT only (~41%) is much higher compared to other studies of general populations. For example, in non-diabetic US adults, IGT only accounted for ~16% of pre-diabetes, (Diabetes Care (2010) 33: 2355–2359). This has important implications, not least that in the absence of an OGTT, two-fifths of the pre-diabetes subjects would have been missed. Also, can the authors discuss possible reasons why the CCS cohort has such a strong IGT component to pre-diabetes incidence compared to the general population?  Have there been any other published observations of pre-diabetes incidence and stratification for heart disease patients and did they show the same IFG/IGT profiles?
  2. From previous revisions, the authors have added additional Tables and other information to the manuscript. While valid and important, this does make the manuscript more cumbersome to read.  The authors need to decide which 5 of the 9 Tables are most important for the reader and put the remaining 4 as Supplementary data. I suggest to also transfer Figures 2 and 3 into the Supplementary data.
  3. The manuscript is generally well written but there are nevertheless a large number of minor grammatical errors throughout (some of which I corrected in the Minor points but it´s likely that there are some more that I missed). It needs to be thoroughly edited for these small mistakes, and the best way is to find a native English speaker to do this.  

 Minor points:

Line 79:  dysglycaemia spelling

Line 88: “…sharing a common protocol and participating in…..”

Lines 92-94: A retrospective identification of patients who were discharged from one of the 14th hospitals who after invitation for a prospective visit were examined in leading centers in 4 geographical regions.

Line 95: “…for at least 6 months?  or  “….up to 6 months?

Lines 99-100 “The prospective visit consisted of an interview with the patient using the EUROASPIRE V questionnaire”.

Line 113:         “..or were taking metformin for other reasons…”

Line 119-120:  “…by a standardized interview questionanaire.”

Table 1 legend: List the units for glucose concentrations mM (mg.dl-1)

Line 124: “….were tested for normality…”

Lines 129-130 “..binary variables were created where prediabetes and diabetes were assigned a value of 1 and normoglycemia  a value of 0.”

Line 133: “..for whom..”

Line 226: “..in Tables 8 and 9..”

Line 300: “..A recent study has ..”

Round 2

Reviewer 2 Report

The authors have adressed my concerns. Thank you.

Author Response

Thank you for helpful comments.